# Cross-reactivity of r*Pvs*48/45, a recombinant *Plasmodium vivax* protein, with plasma from *Plasmodium falciparum* endemic areas of Africa

**Saidou Balam[1]\*, Kazutoyo Miura[2], Imen Ayadi[3], Drissa Konaté[1], Nathan C. Incandela[3¤a], Valentina Agnolon[4¤b], Merepen A. Guindo[1], Seidina A. S. Diakité[1], Sope Olugbile[3¤c], Issa Nebie[5], Sonia M. Herrera[6], Carole Long[2], Andrey V. Kajava[7], Mahamadou Diakité[1], Giampietro Corradin[3], Socrates Herrera[6,8], Myriam Arevalo Herrera[8]**

**1** Mali International Center for Excellence in Research, University of Sciences, Techniques and Technologies of Bamako, Bamako, Mali, **2** Laboratory of Malaria and Vector Research, National Institute of Allergy and Infectious Diseases, National Institutes of Health, Rockville, Maryland, United States of America, **3** Immunobiology Department, University of Lausanne, Lausanne, Switzerland, **4** Division of Immunology and Allergy, Centre Hospitalier Universitaire Vaudois, Lausanne, Switzerland, **5** Groupe de Recherche Action Santé, Burkina Faso, West Africa, **6** Caucaseco Scientific Research Center, Cali, Colombia, **7** Montpellier Cell Biology Research Center (CRBM), University of Montpellier, Montpellier, France, **8** Malaria Vaccine and Drug Development Center, Cali, Colombia

¤a Current address: Department of Chemistry and Biochemistry, University of California, Los Angeles, California, United States of America
¤b Current address: Labcorp, Meyrin, Switzerland
¤c Current address: Piedmont Oncology Institute, Atlanta, Georgia, United States of America
\* Saidou.balam@gmail.com, Saidou.balam@icermali.org

## Abstract

### Background

*Ps*48/45, a *Plasmodium* gametocyte surface protein, is a promising candidate for malaria transmission-blocking (TB) vaccine. Due to its relevance for a multispecies vaccine, we explored the cross-reactivity and TB activity of a recombinant *P. vivax Ps*48/45 protein (r*Pvs*48/45) with plasma from *P. falciparum*-exposed African donors.

### Methods

r*Pvs*48/45 was produced in Chinese hamster ovary cell lines and tested by ELISA for cross-reactivity with plasma from Burkina Faso, Tanzania, Mali, and Nigeria. In addition, BALB/c mice were immunized with the r*Pvs*48/45 protein formulated in Montanide ISA-51 and inoculated with a crude extract of *P. falciparum* NF-54 gametocytes to evaluate the parasite-boosting effect on r*Pvs*48/45 antibody titers. Specific anti-r*Pvs*48/45 IgG purified from African plasma was used to evaluate the *ex vivo* TB activity on *P. falciparum,* using standard mosquito membrane feeding assays (SMFA).

### Results

r*Pvs*48/45 protein showed cross-reactivity with plasma of individuals from all four African countries, in proportions ranging from 94% (Tanzania) to 40% (Nigeria). Also, the level of

---

**Data availability statement:** The raw data supporting the conclusions of this article will be made available by the authors without undue reservation. All data are in the manuscript and/or Supporting information files.

**Funding:** This study was sponsored by NIH/NIAID 1R01AI121237-01 and partly by the Intramural Research Program of NIAID, NIH. The funders had no role in study design, data collection and analysis, decision to publish, or manuscript preparation.

**Competing interests:** No competing interests.

cross-reactive antibodies varied significantly between countries ($p < 0.0001$), with a higher antibody level in Mali and the lowest in Nigeria. In addition, antibody levels were higher in adults ($\geq 17$ years) than young children ($\leq 5$ years) in both Mali and Tanzania, with a higher proportion of responders in adults (90%) than in children (61%) ($p < 0.0001$) in Mali, where male (75%) and female (80%) displayed similar antibody responses. Furthermore, immunization of mice with *P. falciparum* gametocytes boosted anti-*Pvs*48/45 antibody responses, recognizing *P. falciparum* gametocytes in indirect immunofluorescence antibody test. Notably, r*Pvs*48/45 affinity-purified African IgG exhibited a TB activity of 61% against *P. falciparum* in SMFA.

## Conclusion

Plasma from African volunteers predominantly exposed to *P. falciparum* cross-recognized the r*Pvs*48/45 protein. This, together with the functional activity of IgG, warrants further studies for the potential development of a *P. vivax* and *P. falciparum* cross-protective TB vaccine.

## 1. Introduction

Malaria is a human parasitic disease caused by *Plasmodium* species, including *Plasmodium falciparum, P. vivax, P. malariae, P. ovale* and *P. knowlesi* [1]. Currently, particular attention is given to controlling *P. falciparum* and *P. vivax*, which are responsible for over 90% of the 263 million malaria cases and 597,000 malaria deaths reported worldwide in 2023. Approximately 94% of the cases and 95% of the deaths occurred in Africa, where *P. falciparum* causes 99% of clinical cases [2,3]. In contrast, the *P.vivax* infection, thought to be absent in African populations due to the high prevalence of Duffy negative population, has been recently reported in several regions, although in a limited number of individuals [4]. In Asia and America, both species are endemic, but *P. vivax* accounts for ~74% of malaria cases [2,3].

Improved access to first-line treatment, rapid diagnostic tests, and vector control measures resulted in a significant decrease (~54%) in malaria cases worldwide between 2000 and 2015 [2,3,5]. However, certain regions have shown a significant increase in malaria clinical cases during the last few years [6] due to the emergence of resistance to first-line anti-malarial drugs [7–10] and mosquito resistance to insecticides [11–13]. Therefore, efforts must be intensified to develop novel tools and strategies, such as vaccines, to strengthen malaria control and elimination. Indeed, the most advanced malaria vaccines include RTS, S and R21, both based on the *P. falciparum* circumsporozoite (CS) protein, which have been recommended by the WHO for widespread use in Africa [14,15].

Although *P. vivax* vaccine research lags behind *P. falciparum*, several vaccine-candidate antigens from different parasite development stages are also being investigated in preclinical and clinical phases, including a *P. vivax* CS synthetic vaccines [16], sexual stage antigens like *Pvs*48/45, *Pvs*25, and *Pvs*230 [17–19], and asexual stages antigens [20,21].

*Pvs*48/45 is an orthologous protein to *P. falciparum* and is currently being investigated as a TB vaccine candidate [22,23]. *Pvs*48/45 genes are highly conserved and display an overall sequence homology of ~56% between *P. falciparum* and *P. vivax* [24–26]. The functional analysis of the Ps*48/45* gene in *P. berghei* established its crucial role in the fertility of male gametes [26,27]. In addition, recent studies have shown that communities endemic with *P. vivax* or *P. falciparum* transmission exhibit high recognition of the *Pvs*48/45 protein and demonstrate TB activity in *ex vivo* DMFA [28]. However, in these studies, human samples were collected in

endemic areas where both *P. vivax* and *P. falciparum* are present with significant proportions and cross-reactivity in humans could not be ascertained. Furthermore, in mouse immunization studies, *Pvs*48/45 displayed potent cross-reactive antibodies to *Pfs*48/45 and showed cross-bosting immune responses between *Pfs*48/45 and *Pvs*48/45 antigens [28,29]. This study assesses the cross-recognition of r*Pvs*48/45 by plasma from different *P. falciparum* endemic regions of Africa, where *P. vivax* transmission was unknown at the time of plasma collection, probably due to its potential minimal prevalence. In addition, the study evaluates cross-reactive antibody responses in mice immunization.

## 2. Materials and methods

### 2.1. Ethics, consent, and permissions

This study was conducted using samples previously collected in different studies. In Mali, samples corresponded to a research protocol on malaria immunity (Protocol # 08-I-N120). For Mali (ML), the approval was obtained from the Ethical Committee (EC) of the Faculty of Medicine, Pharmacology and Odonto-Stomatology (FMPOS), University of Bamako, Mali (N°0840/FMPOS). All plasma samples were anonymized, archived and stored at -80°C and used in previous studies [30–34]. For Burkina Faso (BF), no authorization was required for a research study in 1998. For Tanzania (TZ), the approval was obtained from the Commission for Science and Technology (UTAFITI NSR/RCA 90). For Nigeria (NIG), the approval was obtained from the Lagos State University Teaching Hospital (LASUTH) ethical review committee [32]. Plasma from healthy Swiss adults was from those who gave informed consent (IC) to participate in malaria vaccine research in 2012 (NCT01605786). Written IC for collecting iRBCs for IFAT and plasma for ELISA were obtained from all adults. Informed assent (IA) was obtained from children in addition to IC from their parents or legal guardians (code CECIV 1506-2017). The animal studies were approved by the Research Ethics Committee of the School of Health, Universidad del Valle (Cali-Colombia) (Code: 031-015).

### 2.2. Human blood samples

Blood samples were collected from adults (≥17-year-old) and children (≤5-year-old) donors living in four different malaria-endemic countries of Africa: Burkina Faso (BF; adults N = 35), Mali (adults N = 62 and children N = 97), Tanzania (TZ; adults N = 83 and children N = 63) and from Nigeria (NIG; adults N = 10). Whole blood was collected in EDTA tubes, and then the plasma was extracted by centrifugation and stored at -80°C before the different tests. Samples from Burkina Faso (BF) and Nigeria (NIG) were collected from donors living in urban settings. In contrast, samples from TZ and Mali were obtained from rural settings where exposure to malaria is potentially higher than in urban settings. Samples from BF were collected in 1998 in the village of Goundry, 30km from Ouagadougou, the capital city. Samples from Mali were collected from 2009 to 2011 in Kenieroba, Bozokin, and Fourda, villages located in the Bancoumana district, 55 km from Bamako, the capital city, and in Dangassa, a village in the Kourouba District, 80 km from Bamako. Samples from TZ were collected from 1982 to 1984 during a large-scale community-based study undertaken in Ifakara (in the Kilombero District in Morogoro). Samples from NIG were collected in March 2007 from donors living in Lagos, southwest. Anonymized plasma from 10 healthy Swiss adults with no history of malaria exposure was used as negative controls.

### 2.3. Mice immunization and sample collection

Mice samples were collected from a study where BALB/c mice were immunized subcutaneously with CHO-r*Pvs*48/45 protein (20 μg), emulsified in Montanide ISA-51, on days

0, 20, and 40, with protein identity confirmed by SDS-PAGE (Excellgene SA, Monthey, Switzerland) as described elsewhere [28,35] (S1 Fig). In this study, when specific anti-CHO r*Pvs*48/45 antibodies had waned to baseline (day 260), a lysate of $5\times10^5$ extract of mix gametocytes (~80% mature forms) from *P. falciparum* NF54 parasite cultures was formulated in Montanide ISA 51 and inoculated intramuscularly (i.m.) to assess the potential boosting effect of *P. falciparum* gametocytes on anti-CHO r*Pvs*48/45 antibody titers. Four weeks after, mice were bled from the submandibular veins (~100 μL), and specific anti-r*Pvs*48/45 antibodies were analyzed by ELISA. Plasma was maintained cryopreserved (-20°C) until use.

## 2.4.  ELISA assays

**Indirect ELISA** was performed using Maxisorp 96-well microtiter plates (Thermo Scientific, Ref. 442404) coated with 50 μL of a 2 μg/mL solution of the r*Pvs*48/45 protein solution overnight at 4°C. The plates were incubated (blocked) for 1 hour at RT with phosphate-buffered saline (PBS) containing 3% non-fat milk powder (PBSx1- milk 3%), then incubated for 2 hours at room temperature (RT) with human plasma at a dilution of 1:200 in PBS containing 3% milk and 0.05% Tween 20 (PBS-T). The plates were then washed four times with PBS-T. Goat anti-human IgG conjugated with horseradish peroxidase (HRP) was used as the secondary antibody at a dilution of 1:2000 (Life technologies, Ref H10307) in PBS-T- milk for 1 hour at RT. After four times washing with PBS-T, the signals were revealed using TMB substrate reagent (BD OptEIA, cat 555214) for 25 min in the dark at RT. The reaction was stopped using 50μL of 1M sulphuric acid (Merck, 1.00731.1000), and the optical density (OD) was measured at both 450 nm and 630 nm wavelength, with the latter used for background correction, using a TECAN Nano Quant Infinit M200 PRO spectrophotometer. ELISA was considered positive when a sample's ODs was higher than the mean OD + 3SD of negative controls (naïve human plasma, NHP from Swiss naïve donors) diluted to 1:200.

 **Self-inhibition ELISA** was performed by incubating plasma from BF (BF40 and BF70) for 1 hour at RT at a dilution of 1:200 with r*Pvs*48/45 at 10-fold serial dilutions from starting at 300 μg/mL before transferring the mixture to plate wells coated with the same r*Pvs*48/45. Plates were then incubated for 30 min at RT, and the reactivity was determined as previously described [33]; each test was performed in duplicate. The percentage of inhibition was thus calculated as (mean of Ab OD with r*Pvs*48/45 protein/mean antibody OD without r*Pvs*48/45 protein) x100.

## 2.5.  Affinity purification of anti-*Pvs*48/45 antibodies

A plasma pool collected from TZ adult donors with high anti-CHO-r*Pvs*48/45 antibody titers was used for IgG purification [33,34]. CNBr-sepharose 4B (Amersham Bioscience AB, Uppsala, Sweden) was activated with 1 mM HCl to prepare the antigen-Sepharose conjugate. Then, 5 mg of the CHO-r*Pvs*48/45 protein was dissolved in 1 mL of coupling buffer (0.1 M NaHCO3 containing 0.5 M NaCl, pH 8.0). Plasma samples were diluted 5-fold with PBS (1x) containing 0.5 M sodium chloride, mixed with the antigen-sepharose conjugate, and stirred O/N gently at 4°C as described elsewhere [33]. The bound antibody was eluted with a glycine solution (0.1 M, pH 2.5). The fractions (F1, F2, F3) were collected in TRIS solution (1 M, pH 8.0) to instantly neutralize the solutions before dialyzing them against phosphate buffer (0.1M, pH 7.0). Each fraction's antibody (IgG) concentration was determined by the absorbance of the solution at 280 nm, as previously described [33,34]. In addition, ELISA was used to determine the recognition of r*Pvs*48/45 by each purified antibody fraction.

## 2.6. Indirect immunofluorescence antibody test (IFAT)

Cross-recognition of *P. vivax* and *P. falciparum* was determined by IFAT using plasma from mice immunized with CHO-r*Pvs*48/45. To this end, *P. vivax* iRBC were obtained from infected patients. White blood cells were separated using a 45% percoll gradient centrifugation at 5,000 rpm for 10 min, and an enriched *P. vivax* gametocytes fraction was obtained [36] and used to prepare 12-well glass microscope slides. For *P. falciparum,* mature gametocytes were obtained from *in vitro* culturing of the *Pf*-NF-54 parasite isolates, which were used to prepare IFAT glass microscope slides as described before [37]. IFAT slides were kept at −70 °C before use. For the IFAT reaction, slides were incubated with a pool of plasma samples (at a 1:200 dilution) obtained from mice immunized with r*Pvs*48/45 and with a pool of control plasma from naïve mice (at a 1:20 dilution) in PBS-Evans blue for 30 min. After PBS washing, slides were incubated with fluorescein isothiocyanate (FITC) conjugated anti-mouse IgG antibody at 1:100 dilution. Slides were examined under an epifluorescence microscope, and antibody titers were determined as the reciprocal of the endpoint dilution that showed positive fluorescence.

## 2.7. Transmission-blocking assays

The functional cross-reactivity of human anti-*Pvs*48/45-specific IgG against *P. falciparum* parasites was evaluated by SMFA, as described previously [18,27,38]. Briefly, 163 μg/mL of the test IgG was mixed with 0.15% - 0.2% stage V gametocytemia of *P. falciparum* NF54 strain and then fed to 3-6 day/old female *Anopheles stephensi* in the presence of human complement. The mosquitoes (N = 20 per sample) were maintained for 8 days and dissected to count the number of oocysts in each midgut. A group of mosquitoes (N = 40 per sample) was fed with normal human or normal mouse Protein-G purified antibodies and used as a negative control.

## 2.8. Statistics

All ELISA data are presented as triplicate wells' average optical density (OD) values. The Mann-Whitney test was utilized to compare two groups, and a Kruskal-Wallis test, followed by Dunn's multiple comparison test, was used to compare more than two groups. A Fischer exact test was used to compare the relative proportion of responding plasma between two groups, and a chi-square test was performed for more than two groups. If the chi-square test shows a significant difference among groups, Fischer's exact test was used to compare two groups simultaneously, and Bonferroni corrected p-values were calculated. GraphPad Prism software, version 5.0, was used for the analysis. A descriptive statistical analysis of median OD, quartile 1 (Q1), and Q3 was used to measure the variation in OD values. The percent reduction of the mean oocyst intensity (TRA) was calculated using the formula: $[(Xc - Xa)/Xc)] \times 100$, where X is the arithmetic mean oocyst intensity in control (c) and test (a) IgG. The 95% confidence interval and *p*-value for TRA (either from a single assay or two assays) were calculated using a zero-inflated negative binomial model as previously described [38].

## 3. Results

### 3.1. Protein purity and cross-reactivity of r*Pvs*48/45 protein with *P. falciparum* adult immune plasma from the four African country donors

After CHO-r*Pvs*48/45 expression, the protein was purified by affinity chromatography, and purity was confirmed by MS/MS [28]. The sequences alignment of full-length *Pvs*48/45 and *Pfs*48/45 proteins confirmed an overall homology of (60.8%) [28], with an even higher homology (> 80%) in the carboxyl region (aa 284-428) (S2 Fig). Plasma samples from BF (adults), Mali (adults and children), TZ (adults and children), and from NIG (adults) were analyzed by

ELISA for their recognition of the CHO-r*Pvs*48/45. Analysis of adult samples indicated a high recognition of CHO-r*Pvs*48/45 proteins in individual plasma across the four participant countries, with a proportion of positive responders ranging from 40-94%. Indeed, the responder's proportion was high in TZ (94%), Mali (90%), and BF (90%), whereas in NIG, the responder's proportion was lower (40%) (p < 0.0001; Fig 1A table). Furthermore, significant variations were observed in OD values among the different countries (p < 0.0001; Fig 1A). We observed that OD values were more similar between adult groups from Mali (with a median OD (Q1; Q3) of 0.310 (0.170; 0.450)) and TZ (with a median (Q; Q3) of 0.252 (0.183; 0.342), but these were slightly higher than BF (with a median OD of 0.205 (0.169; 0.302)) which in turn, was significantly higher (p < 0.0001) than NIG with a median OD of 0.045 (0.033; 0.086).

To further characterize the cross-reactive binding of African antibodies toward r*Pvs*48/45, the r*Pvs*48/45 protein was inhibited with itself (Fig 1B). ELISA was performed using the best two responder plasma from BF adult donors, BF40 and BF70. Inhibition of anti-r*Pvs*48/45 antibody binding to r*Pvs*48/45 protein adsorbed on the ELISA plate was ~ 80% for both plasma at 300 µg/mL.

### 3.2.  Age- and gender-dependent cross-reactivity of the r*Pvs*48/45 protein with *P. falciparum* African immune plasma

Moreover, we analyzed the cross-reactivity of r*Pvs*48/45 against the individual plasma related to the age of donors (adult *vs.* children) in Mali and TZ. In Mali, adults demonstrated a

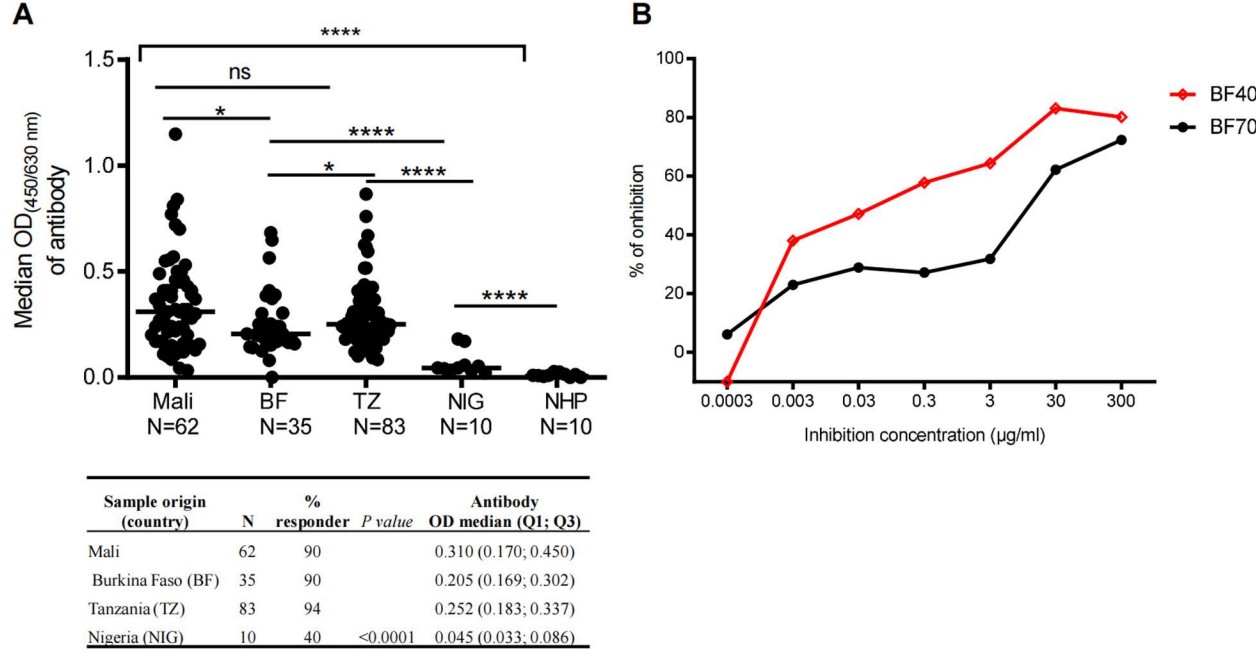

| Sample origin (country) | N | % responder | P value | Antibody OD median (Q1; Q3) |
|---|---|---|---|---|
| Mali | 62 | 90 | | 0.310 (0.170; 0.450) |
| Burkina Faso (BF) | 35 | 90 | | 0.205 (0.169; 0.302) |
| Tanzania (TZ) | 83 | 94 | | 0.252 (0.183; 0.337) |
| Nigeria (NIG) | 10 | 40 | <0.0001 | 0.045 (0.033; 0.086) |

**Fig 1.  Distribution of cross-reactive antibody responses against r*Pvs*48/45 in *P. falciparum* naturally exposed populations from different African endemic areas.** Adult plasma from Mali, Burkina Faso (BF), Tanzania (TZ) and Nigeria (NIG) were collected and tested by ELISA against r*Pvs*48/45 at a dilution of 1/200. Swiss donors' naïve human plasma (NHP) was used as a negative control. **A)** Global analysis of samples shows antibody levels (median OD shown as a horizontal black line in the dot plots) for r*Pvs*48/45 among Mali, Tanzania (TZ), Burkina Faso (BF) and Nigeria (NIG). The inserted table shows the proportion of responder samples among the countries and the median OD, quartile 1 (Q1) and quartile 3 (Q3) of antibody responses. **B)** Two adult BF plasma (BF40 and BF70) with best responses in A were used to inhibit r*Pvs*48/45 coated in an ELISA plate with itself (inhibitor) mixed with prior the plasma samples at dilutions of 1:400 and 1:200, respectively. Dilutions of 1:400 and 1:200 represent 50% of the maximum indirect ELISA signal for each sample. The chi-square test was used to compare the proportions of responder samples. Kruskal-Wallis test, followed by Dunn's multiple comparison tests, was applied to compare the OD median of antibodies among the groups. Naïve human plasma (NHP) was used as the negative control. ****p ≤ 0.0001; ns; not significant; N, number total of samples (plasma donors).

significantly higher responder proportion (p < 0.0001) to r*Pvs*48/45 and showed a drastically higher median OD of 0.310 (0.170; 0.450) than children (0.151 (0.100; 0.245) (Fig 2A). For individual plasma from TZ, adults and children presented comparable responder rates (p > 0.05) and comparable antibody levels with median OD of 0.252 (0.183; 0.342) and 0.224 (0.176; 0.305) (p < 0.05; Fig 2A). However, plasma from children in TZ showed a significantly higher responder proportion (p < 0.001) and higher OD (p < 0.0001) than those from Mali. In contrast, the responder proportion and antibody levels remained comparable in adult donors between the two countries (Fig 2A). Of the 97 samples collected from Malian children, gender information was available only for 41 samples; thus, we evaluated the gender effect for them. No significant difference was observed in the antibody responder proportion between male (75%; N = 16) and female (80%; N = 25) young children from Mali (Fig 2B attached table), nor in the antibody level between these two groups (Fig 2B).

### 3.3. Functional cross-reactivity of r*Pvs*48/45-specific African IgG against *P. falciparum* parasites

Twelve adult samples from Tanzania (TZ) with the highest response against r*Pvs*48/45 (Fig 1 and 2A) were further screened at serial dilutions by ELISA to select and pool three of the

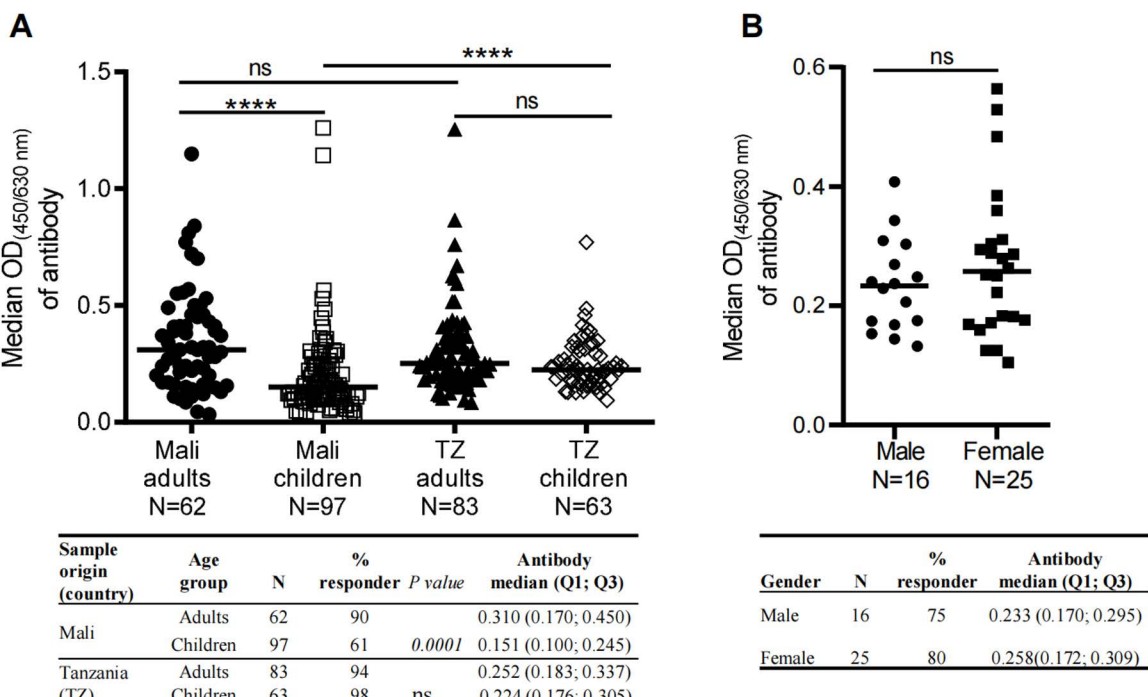

| Sample origin (country) | Age group | N | % responder | *P value* | Antibody median (Q1; Q3) |
|---|---|---|---|---|---|
| Mali | Adults | 62 | 90 | | 0.310 (0.170; 0.450) |
| | Children | 97 | 61 | 0.0001 | 0.151 (0.100; 0.245) |
| Tanzania (TZ) | Adults | 83 | 94 | | 0.252 (0.183; 0.337) |
| | Children | 63 | 98 | ns | 0.224 (0.176; 0.305) |

| Gender | N | % responder | Antibody median (Q1; Q3) |
|---|---|---|---|
| Male | 16 | 75 | 0.233 (0.170; 0.295) |
| Female | 25 | 80 | 0.258 (0.172; 0.309) |

**Fig 2. Distribution of cross-reactive antibodies against r*Pvs*48/45 with regards to the age and gender.** In addition to adult samples from Mali and TZ, as described in Fig 1, ELISA tested additional plasma from young children (≤ 5 years) for rPvs48/45 protein recognition at 1:200. **A)** Levels of cross-reactive antibodies (median OD shown as a horizontal black line in the dot plots) between adults and children in Mali and TZ. The table shows the proportion of responders against r*Pvs*48/45 in adult and child groups in the two countries. The table also shows the median, quartile 1 (Q1), and quartile 3 (Q3) OD's of antibodies against r*Pvs*48/45 protein. The chi-square test was used to compare the responder proportions, and the Kruskal-Wallis test, followed by Dunn's multiple comparison tests, was applied to compare the OD median of the antibody. ***p ≤ 0.001; ****p ≤ 0.0001; ns; not significant; N, number total of donors. **B)** Analysis of the cross-reactive antibody responses for r*Pvs*48/45 in young children (≤ 5 years old) from Mali according to gender (male and female). **A)** The inserted table shows the proportion of responder samples for r*Pvs*48/45 between males and females. Fisher's exact test was performed to compare the proportion of responders between the two genders. The Mann-Whitney test was applied to compare the variation of OD's between males and females. N, the total number of males or females; ns: not significant.

samples with the most robust responses for specific IgG purification against rPvs48/45 (data not shown). Fraction 1 (F1) of the purified IgG, which resulted in the highest antibody concentration, was then used in SMFA for functional cross-reactivity against *P. falciparum* gametocytes (Table 1). The purified rPvs48/45-specific IgG showed a 61% inhibition of *P falciparum* oocyst intensity (95% CI, 31 to 79%; p = 0.003) at 163 μg/mL (the highest concentration used based on the available IgG). DMFA with *P. vivax* could not be performed due to the unavailability of affinity purified antibodies from African plasma, which were denatured because of prolonged storage.

### 3.4. Mice immunization with *P. falciparum* gametocytes boosts anti-*Pvs*48/45 antibody responses, which recognize *P. falciparum* gametocytes in IFAT

In ELISA, a strong anti-CHO-rPvs484/45 antibody response was observed in BALB/c mice that were immunized with CHO-rPvs484/45 protein (round symbols) on days 0, 20 and 40 (thin arrows) and no seroconversion was observed in the control group (square symbols). The antibody responses of the experimental group declined subsequently and reached baseline levels on day 260. Further immunization with a single dose of *P. falciparum* gametocytes (5 x10⁵) significantly boosted anti rPvs48/45 ELISA antibody titers (Fig 3A). Additionally, IFAT using the pooled plasma collected on day 320 (60 after *P. falciparum* boost) showed strong reactivity with enriched *P. falciparum* gametocyte preparation. Additionally, IFAT using pooled plasma collected on day 320 (60 days after the *P. falciparum* boost) demonstrated strong reactivity with the enriched *P. falciparum* gametocyte preparation, potentially reflecting also primary IgM responses to other *P.falciparum* gametocyte proteins, although this was not specifically tested. (Fig 3B, bottom panel). The pooled plasma collected on day 120 (before *P. falciparum* boost) reacted strongly with *P. vivax* homologous antigens (Fig 3B, upper panel).

## 4. Discussion

*Pvs*48/45 is a protein expressed on the surface of *P. vivax* gametocytes, known to be involved in parasite fertilization [26,27]. Both *P. falciparum* and *P. vivax* 48/45 proteins are well-established as targets of natural antibody responses to parasitic sexual stages, which have shown important TB activity in *ex vivo* assays. Consequently, they are currently being pursued as TB vaccine candidates [39–43].

This study indicates that plasma from a significant proportion of donors (40-94%) living in *P. falciparum*-endemic areas of Africa cross-recognize the rPvs48/45 protein. The high and consistent recognition of rPvs48/45 by plasma from different endemic regions of Africa, with no *P. vivax* transmission at the time of plasma collection, confirms previously reported cross-reactivity [44]. Despite the recent confirmation of *P.vivax* transmission in other areas of Africa,

**Table 1. Transmission-blocking activity of anti-rPvs48/45-specific IgG from African donors.**

| Sample name | 1st SMFA | | 2nd SMFA | | % inhibition | 95% CI[d] | p-value |
|---|---|---|---|---|---|---|---|
| | Mean oocyst | Mosquitoes[c] | Mean oocyst | Mosquitoes[c] | | | |
| *Control IgG*[a] | 9.1 | 36/40 | 15.3 | 31/40 | | | |
| *Anti-rPvs48/45 IgG*[b] | 4.3 | 13/20 | 4.9 | 11/20 | 61.2 | 31.2 to 78.6 | 0.003 |

[a]In the first assay, normal human IgG at 3750 μg/mL was used as a control and normal mouse IgG at 750 μg/mL was used in the second assay.

[b]The anti-rPvs48/45-specific IgG from Tanzania adults was tested at 163 μg/mL in both assays.

[c]Number of infected mosquitoes (mosqs)/ Number of dissected mosquitoes.

[d]95% confidence interval.

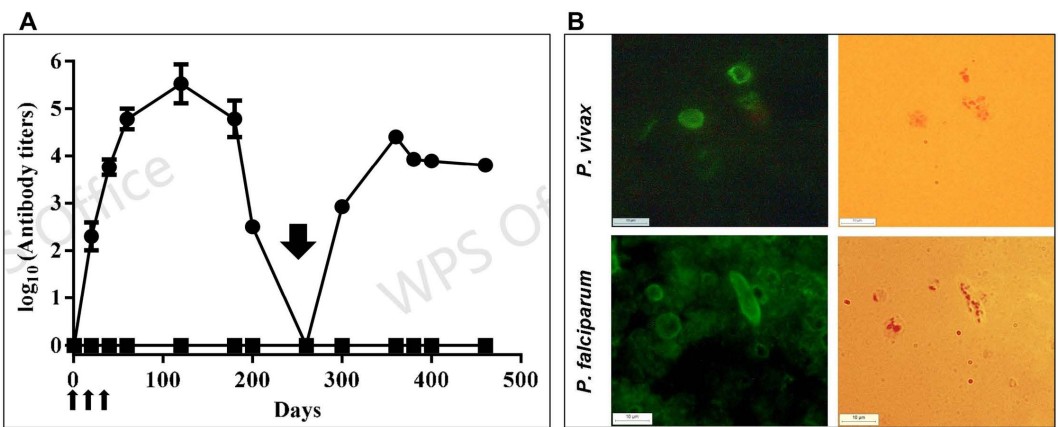

**Fig 3. Mice immunized with *P. falciparum* gametocytes boost the anti-*Pvs*48/45 antibody responses recognizing *P. falciparum* gametocytes in IFAT. A)** Groups of experimental and control mice were immunized on days 0, 20, and 40 (thin arrows) with r*Pvs*48/45 protein (round symbol) or placebo (square symbol), respectively, then boosted with 5 x 10⁵ *P. falciparum* NF-54 gametocytes emulsified in Montanide ISA-51 on day 260 (bold arrow). ELISA determined the antibody titers in individual mice at different time points. Mean and standard deviation in log-transformed antibody titers are shown. **B)** A pooled mouse plasma collected on day 120 (before *P. falciparum* boost) was tested by IFAT with P. vivax (upper panel), and another pooled plasma collected on day 320 (60 after P. falciparum boost) was tested with *P. falciparum g*ametocytes (bottom panel). Parasites were seen under light (left) or epifluorescence (right) microscopy with a 100X objective lens are shown. Picture scale 238 μm.

its prevalence appears minimal [4,44]. Moreover, previous studies in endemic areas of Colombia (Latin America), where both parasites coexist, suggested cross-boosting of the antibody responses in natural conditions as anti-*Pvs*48/45 were higher in regions (Tumaco, Nariño) with higher prevalence of *P.falciparum* [28], which encouraged this study. Moreover, mice immunization studies have unequivocally confirmed this cross-reactivity [23] the present study more demonstrates the cross-reactivity of *Pvs*48/45 against samples from *P. falciparum*-endemic areas. This cross-reactivity is most likely explained by the significant amino-acid sequence homology ( ~ 60,8%) between *P. vivax* and *P. falciparum Ps*48/45 [24,25]. This feature makes this protein a promising target for developing an effective cross-species vaccine.

The communities analyzed in this study have been historically exposed to variable *P. falciparum* transmission intensities [3,8] and minimal *P.vivax* prevalence [4]. The r*Pvs*48/45 recognition intensity, as determined by the level of specific anti-r*Pvs*48/45 and the percentage of positive responders in each of the four endemic countries, may be correlated with the relative transmission intensity and episodes history of malaria in these countries. Further research is now necessary to test these hypotheses. Furthermore, despite a high proportion of positive serological responses, our research highlights significant differences in antibody levels (OD value) based on demographics. While cross-reactive antibody levels were more comparable and higher for Mali and TZ where samples were collected from rural sites, than those from BF or NIG, collected from urban sites. These findings support the argument that populations living in rural communities and small villages are more likely to be exposed to malaria vectors than those living in metropolitan areas [45].

In addition to location, age played an important role in antibody response. Adults comprised a greater proportion of responders and demonstrated higher antibody levels than children, mainly in Mali. This suggests that cross-reactive immune responses to the r*Pvs*48/45 protein may increase with age, which is argued to be a result of adults being exposed repeatedly. Hence, these cross-reactive anti-r*Pvs*48/45 antibodies are likely acquired early in childhood and are then boosted throughout the donors' lives, most likely in response to subsequent

*P. falciparum* infections. This age-related increasing trend of the specific immune response in the naturally exposed population has also been demonstrated with other malaria antigens. However, such immunity was due to antigens specific to *Plasmodium* species [20,41,46–48]. Moreover, recent seroepidemiological studies performed with plasma from malaria-endemic regions where both *P. vivax* and *P. falciparum* are co-transmitted suggest that frequent exposure to *P. falciparum* infections results in anti-*P. vivax* antibodies maintenance [23,28,40,42]. However, this study did not determine which Ps48/45 segments were cross-reactive nor demonstrated which conserved protein domains are prone to be restricted despite recognizing heterologous parasites observed [28,49,50]. One of the limitations of this study is that these plasma were not tested with the recombinant *Pf*s48/45 protein. However, in Africa, where *Pf* is present, it has been shown that a naturally acquired antibody response to the recombinant *Pf*s 48/45 protein is detected [51–53]. Another limitation of the study is that the gender effect was assessed only in a part of Malian children samples, where gender information was available. Further study is required to determine whether there is no gender difference in other populations.

Altogether, the consistent reactivity of antibodies against r*Pvs*48/45 protein in both *P. falciparum* and *P. vivax* parasites under natural conditions in distant continents with significant epidemiological differences and in animal models correlates with the protein sequence conservation. The results of populations naturally exposed to *P. falciparum* are also in agreement with the cross-reactivity and cross-boosting effects observed in mice experimentally immunized with r*Pfs*48/45 and r*Pvs*48/45 *E. coli* recombinant products [28,29]. Moreover, the anti-r*Pvs*48/45 response appears consistent with the high proportion of antibodies against *Pvs*48/45 reported in adults from malaria-endemic areas of Colombia (Latin America), where malaria transmission is significantly lower [28,54].

More importantly, there was a significant *ex-vivo* reduction of 61.2% of *P. falciparum* oocyst development in *An. stephensi* fed with *P. falciparum* gametocytes by affinity-purified anti-r*Pvs*48/45 IgG is of interest and encourages investing further efforts into characterizing the functional domains to model multispecies TB vaccine development. Although the cross-species *ex vivo* TB activity is suboptimal, the likelihood of inducing robust TB in both species through vaccination is highly likely.

The recognition of the native proteins of the two species in IFAT assays, the *P. falciparum* *ex-vivo* TB activity and the *P. falciparum* gametocyte boosting of anti-CHO r*Pvs*48/45 antibodies in mice provide further support for r*Pvs*48/45 as a target for a TB vaccine. This *P. falciparum* boosting may have induced antibody responses to other gametocyte proteins, which were not specifically tested. Moreover, the current epidemiological data, together with the ELISA cross-species reactivity and the TB capacity of r*Pvs*48/45-specific antibodies purified from *P. falciparum* semi-immune individuals against, support the further development of a cross-species TB vaccine.

## Supporting information

**S1 Fig. Recombinant CHO-rPvs48/45 protein analysis in western Blott.** CHO-rPvs48/45 protein identity was confirmed using 12% SDS-PAGE gel in western blot. Analysis was carried out under reducing (0.05 mol/L dithiothreitol, DTT) and non-reducing conditions (wt) [22]. (TIFF)

**S2 Fig. Sequence homology between the Pvs48/45 and Pfs48/45 proteins.** The amino acid (aa) sequences alignment of full-length Pvs48/45 and Pfs48/45 proteins were obtained using the PlasmoDB database, and sequences matched with Blastp (protein-protein BLAST; https://bit.ly/3C0hPpK). Pfs48/45 and Pvs48/45 share ~56% identity (238 out of 423) and ~ 78%

similarity in their protein sequences. Conserved cysteine residues are identified by the C letter in black and bold. In the red sequence, identical amino acid residues are identified by (letter|), the similar residues by (+) and the different residues by (*).
(TIFF)

**S1 Checklist. Humane endpoints checklist.**
(DOCX)

## Acknowledgments

We are grateful for the participation of the community from malaria-endemic countries of Mali, Tanzania, Burkina Faso, Nigeria, and Swiss volunteers. We want to thank Drs. Marcel Tanner and Ingrid Felger for sharing plasma from malaria-endemic areas and acknowledging the intramural program of the National Institute of Allergy and Infectious Disease.

## Author contributions

**Conceptualization:** Saidou Balam, Sope Olugbile, Giampietro Corradin, Socrates Herrera, Myriam Arevalo Herrera.

**Data curation:** Saidou Balam.

**Formal analysis:** Saidou Balam, Kazutoyo Miura, Imen Ayadi, Drissa Konaté, Carole Long, Andrey V. Kajava.

**Funding acquisition:** Myriam Arevalo Herrera.

**Investigation:** Saidou Balam, Socrates Herrera, Myriam Arevalo Herrera.

**Methodology:** Kazutoyo Miura, Imen Ayadi, Nathan C. Incandela, Valentina Agnolon, Merepen A Guindo, Seidina A.S. Diakité, Issa Nebie, Sonia M Herrera, Carole Long, Andrey V. Kajava, Mahamadou Diakité.

**Resources:** Saidou Balam.

**Supervision:** Saidou Balam, Giampietro Corradin, Socrates Herrera.

**Validation:** Saidou Balam, Socrates Herrera, Myriam Arevalo Herrera.

**Visualization:** Saidou Balam, Socrates Herrera.

**Writing – original draft:** Saidou Balam, Drissa Konaté, Sonia M Herrera, Giampietro Corradin, Socrates Herrera.

**Writing – review & editing:** Saidou Balam, Sonia M Herrera, Giampietro Corradin, Socrates Herrera, Myriam Arevalo Herrera.

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
