## [Decision Letter · Decision Letter 0]

11 Sep 2024

PONE-D-24-12287Cross-reactivity of rPvs48/45, a recombinant Plasmodium vivax protein, with sera from Plasmodium falciparum endemic areas of AfricaPLOS ONE

Dear Dr. Balam,

Thank you for submitting your manuscript to PLOS ONE. After careful consideration, we feel that it has merit but does not fully meet PLOS ONE’s publication criteria as it currently stands. Therefore, we invite you to submit a revised version of the manuscript that addresses the points raised during the review process.

Please carefully respond to all comments from both of the reviewers, and modify your manuscript accordingly.

We look forward to receiving your revised manuscript.

Kind regards,

David Joseph Diemert, M.D.

Academic Editor

PLOS ONE

“MAH has received the award.

This study was sponsored by National Institute of Health NIH/NIAID 1R01AI121237-01 and in part by the Intramural Research Program of NIAID, NIH.”

Reviewers' comments:

Reviewer's Responses to Questions

**Comments to the Author**

1. Is the manuscript technically sound, and do the data support the conclusions?

Reviewer #1: Yes

Reviewer #2: Yes

2. Has the statistical analysis been performed appropriately and rigorously? 

Reviewer #1: Yes

Reviewer #2: Yes

3. Have the authors made all data underlying the findings in their manuscript fully available?

Reviewer #1: Yes

Reviewer #2: Yes

4. Is the manuscript presented in an intelligible fashion and written in standard English?

Reviewer #1: Yes

Reviewer #2: Yes

5. Review Comments to the Author

Reviewer #1: This manuscript reports essentially confirmatory data on the recognition of Pvs48/45 by antibodies in people previously exposed to P. falciparum thus confirming cross-reactivity on a larger sample size from 4 different African countries. The authors have cited previously published work on the same phenomenon of cross-reactivity. The work in the cited reference showed >70% cross recognition by sera from a study conducted in Southern African region (Zimbabwe). While the studies and data support the conclusions drawn, I would like to see authors respond to my major and minor concerns listed below:

1. Line 61:replace ...,"(exposed only to P. falciparum)".... with ....."exposed predominantly to P. falciparum"...., unless authors have tested blood for infecting parasites. Co-infections with other Plasmodium species are quite common.

2. Lines 83-97: should be deleted because they are irrelevant to the studies.

3. The Introduction is vey unfocussed and very similar superfluous and needs to be carefully made succinct with the overall goal of studies properly stated with due credit to previously published work.

4. 2.1 and 2.2 have very similar details and efforts need to be made to minimize duplication.

5. 2.3: Do not include results in the methods.

6. 2.4 sub heading can be deleted and details can simply be stated because they have been published previously and well known at this point.

7. Similarly 2.5 details are likely published in 62 and be referenced without the need for a sub-heading.

8. Line 77: what is the superscript 2 on the word derivatives for?

9. 2.6: Why have authors included Ref number for each chemical from a particular vendor?

10. This reviewer does not like the term competition term for the ELISA. All they are doing is to self deplete cross-reacting antibodies. In the same they include ref. 72. Are any details different from those in the indirect ELISA?

11. 2.9: Why did they include complement in the SMFA? Does it depend upon complement?

12. Line 361: the reference 81 pointed out above was from a study on sera from Africa and not Latin America and that study too showed >70% sera positive for cross-reactivity.

13. The studies on P. falciparum SMFA with affinity purified are nice and not surprising. What the authors must do is to evaluate the same antibody in DMFA with P. vivax. In the absence of this data the present studies are only confirmatory and add incremental knowledge.

14: Comments on Figures:

Figures 1,2 and 3: Delete Panel A and simply add % responded column in the table. In fact I would suggest that modified Figures 1-3 can simply be three panels of a single Figure.

Figure 4: It can be deleted because it does not add any knowledge. Panel A simply shows that BF40 has less antibody than BF 70. Panel B ELISA titration curves for a few select sera. Panel C just shows which eluted fraction has maximum eluted antibody amount. The entire figure does not contribute anything new.

Figure 5: The authors need to demonstrate IgM (day 270-280) and IgG (days >300) responses against Pfs48/45 after boosting with parasite lysates. The overall IFA quality was unacceptably poor.

Reviewer #2: This is an interesting report suggesting cross-species protection by a Plasmodium vivax antigenic protein possibly involved in transmission-blocking activity against P. falciparum.

There is one major concern with the interpretation offered. The authors confidently assert that the African sera examined would have never been exposed to any P. vivax malaria at all. This view seems to hinge on the sera being collected prior to the now well known and commonly reported presence of endemic P. vivax in all of the nations represented in their samples. This presumes the flurry of P. vivax reports out of Africa starting in about 2010 documented a new invasion of the continent and that prior to this, no P. vivax transmission occurred. This is very highly unlikely. Instead, P. vivax transmission has very probably always occurred in Africa but went unnoticed due to inherently very low parasitemias ,or due to microscopic misidentification as P. ovale. This reviewer does not accept the presumption of cross-reactive P. vivax antigen in people who very probably had experience with P. vivax infection.

The authors should consider the likelihood of the observed seropositivity being due to actual exposure to endemic P. vivax.

6. PLOS authors have the option to publish the peer review history of their article (what does this mean? ). If published, this will include your full peer review and any attached files.

**Do you want your identity to be public for this peer review?** For information about this choice, including consent withdrawal, please see our Privacy Policy .

Reviewer #1: No

Reviewer #2: No

---

## [Author Response · Author response to Decision Letter 1]

29 Nov 2024

Responses to Review Comments

Reviewer #1:

This manuscript reports essentially confirmatory data on the recognition of Pvs48/45 by antibodies in people previously exposed to P. falciparum thus confirming cross-reactivity on a larger sample size from 4 different African countries. The authors have cited previously published work on the same phenomenon of cross-reactivity. The work in the cited reference showed >70% cross recognition by sera from a study conducted in Southern African region (Zimbabwe). While the studies and data support the conclusions drawn, I would like to see authors respond to my major and minor concerns listed below:

Comment # 1. Line 61:replace ...,"(exposed only to P. falciparum)".... with ....."exposed predominantly to P. falciparum"...., unless authors have tested blood for infecting parasites. Co-infections with other Plasmodium species are quite common.

Response #1: The likelihood of having P. vivax circulation in the study areas was mentioned in the summary, introduction and discussion sections.

Comment # 2. Lines 83-97: should be deleted because they are irrelevant to the studies.

Response # 2: We have done extensive editing between lines 77-108

Comment # 3. The Introduction is very unfocussed and very similar superfluous and needs to be carefully made succinct with the overall goal of studies properly stated with due credit to previously published work.

Response #3. The introduction section has been revised to enhance focus and conciseness and adequately establish the study’s objectives in light of previous research.

Comment # 4. 2.1 and 2.2 have very similar details and efforts need to be made to minimize duplication.

Response #4. We thank the reviewer for this remark. We have reviewed the sub-chapters and eliminated redundancies.

Comment # 5. 2.3: Do not include results in the methods.

Response # 5: The section now exclusively describes the immunization of mice and sample collection methods, with results moved to the corresponding section.

Comment # 6. 2.4 subheading can be deleted, and details can simply be stated because they have been published previously and well known at this point.

Response # 6: This section has been removed, and references to previous studies have been cited in the relevant section of the Results or Discussion.

Comment # 7. Similarly, 2.5 details are likely published in 62 and be referenced without the need for a sub-heading.

Response # 7. We have removed the subheading and cited the referenced work.

Comment # 8. Line 77: what is the superscript 2 on the word derivatives for?

Response #8. This typo error was corrected.

Comment # 9. 2.6: Why have authors included Ref number for each chemical from a particular vendor?

Response # 9: These references were added to ensure precision, but we acknowledge the potential for redundancy, and they are removed.

Comment # 10. This reviewer does not like the term competition term for the ELISA. All they are doing is to self deplete cross-reacting antibodies. In the same they include ref. 72. Are any details different from those in the indirect ELISA?

Response # 10. We appreciate the feedback. This assay is indeed more accurately described as an auto-inhibition ELISA to deplete cross-reacting antibodies against rPvs48/45. This clarification has been made in the Methods section under the ELISA description.

Comment # 11. 2.9: Why did they include complement in the SMFA? Does it depend upon complement?

Response #11. Complement was included based on its role in immune-mediated parasite killing, particularly within transmission-blocking immunity. Complement can enhance antibody efficacy against Plasmodium parasites by facilitating opsonization, membrane attack complex formation, and phagocytosis. Although the SMFA can operate without complement, we included it to explore if it might enhance antibodies' effectiveness against rPvs48/45. While complement is not essential for the assay, this approach allowed us to assess potential synergistic effects on antibody neutralization.

Comment # 12. Line 361: the reference 81 pointed out above was from a study on sera from Africa and not Latin America and that study too showed >70% sera positive for cross-reactivity.

Response #12. The original MS had only 70 references. Anyway, the reference to studies in Latin America, specifically in Colombia, was revised and explained further.

Comment # 13. The studies on P. falciparum SMFA with affinity purified are nice and not surprising. What the authors must do is to evaluate the same antibody in DMFA with P. vivax. In the absence of this data the present studies are only confirmatory and add incremental knowledge.

Response #13. We appreciate the reviewer's suggestion and agree that conducting DMFA with P. vivax would provide valuable additional data. Unfortunately, the antibody that had been affinity-purified from African sera in Switzerland became denatured due to long storage in customs and could not be used. We are exploring possibilities for conducting DMFA with P. vivax in future studies.

Comment # 14: Comments on Figures:

Figures 1,2 and 3: Delete Panel A and simply add % responded column in the table. In fact, I would suggest that modified Figures 1-3 can simply be three panels of a single Figure.

Figure 4: It can be deleted because it does not add any knowledge. Panel A simply shows that BF40 has less antibody than BF 70. Panel B ELISA titration curves for a few select sera. Panel C just shows which eluted fraction has the maximum eluted antibody amount. The entire figure does not contribute anything new.

Figure 5: The authors need to demonstrate IgM (day 270-280) and IgG (days >300) responses against Pfs48/45 after boosting with parasite lysates. The overall IFA quality was unacceptably poor.

Response #14: We thank the reviewer for this insightful suggestion. Panels A of Figures 1, 2, and 3 have been removed, and the percentage of respondents has been added to the associated table. This further illustrates that the rPvs48/45 protein's conformation does not differ or change between the liquid and solid phases in the ELISA plate. Further details are also provided in the Results section page 12 lines 250-255 (MS Revised clean version).

Figure 4 comment was deleted.

Figure 5: IgG is included in current Fig 3. Specific IgM was not detectable at these dates.

Reviewer #2:

This is an interesting report suggesting cross-species protection by a Plasmodium vivax antigenic protein possibly involved in transmission-blocking activity against P. falciparum.

There is one major concern with the interpretation offered. The authors confidently assert that the African sera examined would have never been exposed to any P. vivax malaria at all. This view seems to hinge on the sera being collected prior to the now well known and commonly reported presence of endemic P. vivax in all of the nations represented in their samples. This presumes the flurry of P. vivax reports out of Africa starting in about 2010 documented a new invasion of the continent and that prior to this, no P. vivax transmission occurred. This is very highly unlikely. Instead, P. vivax transmission has very probably always occurred in Africa but went unnoticed due to inherently very low parasitemias ,or due to microscopic misidentification as P. ovale. This reviewer does not accept the presumption of cross-reactive P. vivax antigen in people who very probably had experience with P. vivax infection.

The authors should consider the likelihood of the observed seropositivity being due to actual exposure to endemic P. vivax.

Response #1. We thank the reviewer for this valuable observation. Although it is uncertain how long P. vivax has been circulating in Africa, it is likely that even if it has been broadly circulating for a long time, it has been very low-density and would not affect the findings of this study. This has been discussed in the current version,

in the conclusion subsection of the Abstract, Introduction and Discussion.

---

## [Decision Letter · Decision Letter 1]

13 Dec 2024

PONE-D-24-12287R1Cross-reactivity of rPvs48/45, a recombinant Plasmodium vivax protein, with sera from Plasmodium falciparum endemic areas of AfricaPLOS ONE

Dear Dr. Balam,

Thank you for submitting your manuscript to PLOS ONE. After careful consideration, we feel that it has been improved and has merit but it still does not fully meet PLOS ONE’s publication criteria as it currently stands. Therefore, we invite you to submit a revised version of the manuscript that addresses the additional points raised during the review process, as outlined below.

We look forward to receiving your revised manuscript.

Kind regards,

David Joseph Diemert, M.D.

Academic Editor

PLOS ONE

Reviewers' comments:

Reviewer's Responses to Questions

**Comments to the Author**

1. If the authors have adequately addressed your comments raised in a previous round of review and you feel that this manuscript is now acceptable for publication, you may indicate that here to bypass the “Comments to the Author” section, enter your conflict of interest statement in the “Confidential to Editor” section, and submit your "Accept" recommendation.

Reviewer #1: All comments have been addressed

2. Is the manuscript technically sound, and do the data support the conclusions?

Reviewer #1: Partly

3. Has the statistical analysis been performed appropriately and rigorously? 

Reviewer #1: Yes

4. Have the authors made all data underlying the findings in their manuscript fully available?

Reviewer #1: Yes

5. Is the manuscript presented in an intelligible fashion and written in standard English?

Reviewer #1: Yes

6. Review Comments to the Author

Reviewer #1: It is obvious that the authors have revised the current manuscript extensively and it is much improved from the previous original submission. Basically the findings confirm previous similar results reported in Ref. 50, however the studies do provide demographically broader cross-reactivity. I suggest following comments for authors to take into consideration to help improve the overall presentation:

1. The Literature should remain focused on the target antigen and the intent should not be to review the literature on P. vivax vaccines. This will focus the manuscript and also help remove unnecessary too many self-citations.

2. Why are there are no co-authors who collected and provided sera from TZ and NIG?

3. line 53: Pf IFA data is questionable because the sera likely contained anti-Pf IgM induced by the Pf booster dose and it may not be due to boosted anti-Pvs48/45 abs.

4. Provide a more recent ref for #3 for WHO

5. Many references are irrelevant (in view of my point 1) . These include 17-19, 23, 25-27.

line 123: it should be plasma because the blood was collected in the presence of EDTA.

6. lines 137-146: I understand these mice data were published previously but the same mice mice after rest period were boosted with Pf. In that case lines 137-146 can be simply replaced with a sentence with appropriate reference.

7. line 165:. Change to "....The reaction was stopped using ...ul 1 M sulfuric acid and the optical density measured.... was the OD measured at both the wavelengths ?

8. Lines 178-183: If these beads were prepared as published then just say so and cite the ref. and delete these lines.

9. Line 253: You don't know whether the conformation was the same or not. You are only looking at the ag-ab reactivity.

10. In Figs. 1 and 2 when presenting human sera OD values - are these observed or after subtraction of (negative control = 3SD)? If these are after subtraction then I don't have any further comment. If not, then the right way will be to present net OD after subtraction. In the latter case all further % values and calculations throughout the manuscript are likely to be affected.

11. sub-heading on lines 274-274: Somehow the use of Concerning is not sitting well in my reading. The authors might want to rephrase.

12. Lines 317-322:Suggest rewrite "...., was then used in SMFA for functional cross-reactivity against Pf gametocytes (Table 1). At the end of this paragraph, the authors should state why they could not those abs against Pv in DMFA.

13. line 338: replace vigorous with strong.

14. ...260. Further immunization with a single dose of Pf gametocytes resulted in significant boosting of anti-Pvs48/45 ELISA....

15. Line 355-357: IFATR concern in view of my comment number number 3.

16. line 369: ....collection, confirms previously reported cross reactivity (ref. 50).

17. Line 375: Move ref. 50 to modified line 369 and ref. 51 is not from mice but humans.

18. Line 397: ...exposed repeatedly. and delete periods as a function of their age from line 398.

19. Line 432: I am not convinced about IFAT data against Pf,as stated before.

20. Line 434: ...in mice provide further support for Pvs48/45 as a target for TB vaccine.

7. PLOS authors have the option to publish the peer review history of their article (what does this mean? ). If published, this will include your full peer review and any attached files.

**Do you want your identity to be public for this peer review?** For information about this choice, including consent withdrawal, please see our Privacy Policy .

Reviewer #1: **Yes: ** Nirbhay Kumar

---

## [Author Response · Author response to Decision Letter 2]

20 Dec 2024

Responses to Reviewer Comments

1. The Literature should remain focused on the target antigen and the intent should not be to review the literature on P. vivax vaccines. This will focus on the manuscript and also help remove unnecessary too many self-citations.

Response (R)- We agree with your suggestion and most of the references not directly related to TB were deleted except those on vaccine necessary for a paragraph description.

2. Why are there are no co-authors who collected and provided sera from TZ and NIG?

R- Dr. Sope Olugbile, co-author of this study, contributed with samples from the NIG. From TZ, nobody contributed to this publication in a manner acceptable for the international standards of authorship.

3. line 53: Pf IFA data is questionable because the sera likely contained anti-Pf IgM induced by the Pf booster dose and it may not be due to boosted anti-Pvs48/45 abs.

R- The data presented reflects the observed results. Given the high homology of P48/45 in both parasite species, is plausible that the immune system generated a secondary IgG response.

4. Provide a more recent ref for #3 for WHO

R- This reference was updated to the 2024 World Malaria Report.

5. Many references are irrelevant (in view of my point 1). These include 17-19, 23, 25-27.

R- Irrelevant references are now removed.

6. Line 123: it should be plasma because the blood was collected in the presence of EDTA.

R- For human samples collected with EDTA the word “sera” was replaced to “plasma” throughout the manuscript text and Figure (s).

7. lines 137-146: I understand these mice data were published previously but the same mice after rest period were boosted with Pf. In that case lines 137-146 can be simply replaced with a sentence with appropriate reference.

R- The description has been now summarized (Cf method-mice immunization section: lines 140-150)

8. Line 165xxx: Change to "....The reaction was stopped using ...ul 1 M sulfuric acid and the optical density measured.... was the OD measured at both the wavelengths ?

R- The description was revised and completed as follow (Lines164-167):

“The reaction was stopped using 50mL of 1M sulphuric acid (Merck, 1.00731.1000) and the optical density (OD) was measured at both 450 nm and 630 nm wavelength, with the later used for background correction, using a TECAN Nano Quant Infinit M200 PRO spectrophotometer”

9. Lines 178-183: If these beads were prepared as published then just say so and cite the ref. and delete these lines.

R- We think it's necessary to maintain a few descriptions of the method. However, the text has been simplified while citing the reference.Lines: 177-190

10. Line 253: You don't know whether the conformation was the same or not. You are only looking at the ag-ab reactivity.

R- You are correct. The sentence was removed (cf. esult section, subtitle a.).

11. In Figs. 1 and 2 when presenting human sera OD values - are these observed or after subtraction of (negative control = 3SD)? If these are after subtraction, then I don't have any further comment. If not, then the right way will be to present net OD after subtraction. In the latter case all further % values and calculations throughout the manuscript are likely to be affected.

R- We would like to thank the reviewer for this point of view. However, it needs to be noted that the ODs presented here are those without subtraction of the ODs of negative control. We believe that since the 3SD of the control are not representing an internal control for each interest sample, it is desirable to present both the ODs of the samples and the negative control as we have shown in figure 1A. Indeed, the OD of the controls is significantly lower than even the OD of the NIG samples, which is the lowest of the four countries (Cf. graphic, Figure 1A). As regards the calculation of prevalence (positive samples), these 3SD of the control are applied to each sample ( cf. ELISA positivity definition, Lines: 167-169).

12. Sub-heading on lines 274-274: Somehow the use of Concerning is not sitting well in my reading. The authors might want to rephrase.

R- We have modified the subtitle as follow in Line 271:

“Age- and gender-dependent cross-reactivity of the rPvs48/45 protein with P. falciparum African immune plasma”

13. Lines 317-322: Suggest rewrite "...., was then used in SMFA for functional cross-reactivity against Pf gametocytes (Table 1). At the end of this paragraph, the authors should state why they could not those abs against Pv in DMFA.

R- The section was rewritten as suggested to add this sentence:” DMFA with P. vivax could not be performed due to the unavailability of affinity purified antibodies from African plasma, which were denatured because of prolonged storage”. Lines 316-318

14. line 338: replace vigorous with strong.}

R- The word was modified. Line 333.

15. ...260. Further immunization with a single dose of Pf gametocytes resulted in significant boosting of anti-Pvs48/45 ELISA....

R- The sentence was modified. Line 337-338.

16. Line 355-357: IFATR concern in view of my comment number 3.

R- We apologize for the error. The same figure was mistakenly included. The IFAT figure has been replaced (Figure 3B).

17. Line 369: ....collection, confirms previously reported cross reactivity (ref. 50).

R- The reference was included. Now is Ref 44 (Bansal GP et al). Line 367.

18. Line 375: Move ref. 50 to modified line 369 and ref. 51 is not from mice but humans.

R- References were modified as suggested Ref 44. Line 367-368. Ref 51. is now not cited as for mouse.

19. Line 397: ...exposed repeatedly. and delete periods as a function of their age from line 398.

R- The sentence was modified as suggested. Line 394.

20. Line 432: I am not convinced about IFAT data against Pf as stated before.

R- IFAT figure was replaced. cf figures 3A.

21. Line 434: ...in mice provide further support for Pvs48/45 as a target for TB vaccine.

R- The sentence was modified. Line 428.

---

## [Decision Letter · Decision Letter 2]

16 Jan 2025

PONE-D-24-12287R2Cross-reactivity of rPvs48/45, a recombinant Plasmodium vivax protein, with plasma from Plasmodium falciparum endemic areas of AfricaPLOS ONE

Dear Dr. Balam,

Thank you for submitting your manuscript to PLOS ONE. After careful consideration, we feel that it has merit but does not fully meet PLOS ONE’s publication criteria as it currently stands. Therefore, we invite you to submit a revised version of the manuscript that addresses the points raised during the review process.

We look forward to receiving your revised manuscript.

Kind regards,

David J. Diemert, M.D.

Academic Editor

PLOS ONE

Journal Requirements:

Reviewers' comments:

Reviewer's Responses to Questions

**Comments to the Author**

1. If the authors have adequately addressed your comments raised in a previous round of review and you feel that this manuscript is now acceptable for publication, you may indicate that here to bypass the “Comments to the Author” section, enter your conflict of interest statement in the “Confidential to Editor” section, and submit your "Accept" recommendation.

Reviewer #1: All comments have been addressed

2. Is the manuscript technically sound, and do the data support the conclusions?

Reviewer #1: Partly

3. Has the statistical analysis been performed appropriately and rigorously? 

Reviewer #1: Yes

4. Have the authors made all data underlying the findings in their manuscript fully available?

Reviewer #1: Yes

5. Is the manuscript presented in an intelligible fashion and written in standard English?

Reviewer #1: Yes

6. Review Comments to the Author

Reviewer #1: All my previous concerns have been addressed except one pertaining to IFA data. I would suggest that authors insert a disclaimer (on lines 340 and 426-427) that the IFA reactivity to Pf may also be due to primary IgM response against immunizing Pf gametoc4yte proteins (not tested).

7. PLOS authors have the option to publish the peer review history of their article (what does this mean? ). If published, this will include your full peer review and any attached files.

**Do you want your identity to be public for this peer review?** For information about this choice, including consent withdrawal, please see our Privacy Policy .

Reviewer #1: **Yes: ** Nirbhay Kumar

---

## [Author Response · Author response to Decision Letter 3]

24 Jan 2025

Reply to Reviewer (s)

**Journal Requirements:

Answer: We have checked the reference list and everything seems to be correct, with no indication of retracted reference.

Reviewer #1: All my previous concerns have been addressed except one pertaining to IFA data. I would suggest that authors insert a disclaimer (on lines 340 and 426-427) that the IFA reactivity to Pf may also be due to primary IgM response against immunizing Pf gametoc4yte proteins (not tested).

Response: We thank the reviewer for the suggestions. We have now revised the relevant paragraphs to address these concerns. Cf. Manuscript with Track changes: Lines 340-344 (results section) and Lines 431-433 (discussion section)

Responses: All figures have now been checked with PACE to meet PLOS recommendations. The figures now supplied are those suggested through PACE.

---

## [Editor Report · Decision Letter 3]

5 Feb 2025

Cross-reactivity of rPvs48/45, a recombinant Plasmodium vivax protein, with plasma from Plasmodium falciparum endemic areas of Africa

PONE-D-24-12287R3

Dear Dr. Balam,

We’re pleased to inform you that your manuscript has been judged scientifically suitable for publication and will be formally accepted for publication once it meets all outstanding technical requirements.

Kind regards,

David J. Diemert, M.D.

Academic Editor

PLOS ONE
---

## [Editor Report · Acceptance letter]

PONE-D-24-12287R3

PLOS ONE

Dear Dr. Balam,

I'm pleased to inform you that your manuscript has been deemed suitable for publication in PLOS ONE. Congratulations! Your manuscript is now being handed over to our production team.

Kind regards,

on behalf of

Dr. David J. Diemert

Academic Editor

PLOS ONE